# Growth, Yield, and Bunch Quality of “Superior Seedless” Vines Grown on Different Rootstocks Change in Response to Salt Stress

**DOI:** 10.3390/plants10102215

**Published:** 2021-10-19

**Authors:** Lo’ay A. A., Dina A. Ghazi, Nadi Awad Al-Harbi, Salem Mesfir Al-Qahtani, Sabry Hassan, Mohamed A. Abdein

**Affiliations:** 1Pomology Department, Faculty of Agriculture, Mansoura University, EL-Mansoura 35516, Egypt; 2Soil Department, Faculty of Agriculture, Mansoura University, EL-Mansoura 35516, Egypt; dinaghazi@mans.edu.eg; 3Biology Department, University College of Tayma, Tabuk University, Tabuk 71411, Saudi Arabia; nalharbi@ut.edu.sa (N.A.A.-H.); salghtani@ut.edu.sa (S.M.A.-Q.); 4Department of Biology, College of Science, Taif University, Taif 21944, Saudi Arabia; hassan@tu.edu.sa; 5Biology Department, Faculty of Arts and Science, Northern Border University, Rafha 91911, Saudi Arabia

**Keywords:** rootstocks, soil salinity, superior seedless, bunches quality

## Abstract

The growth and quality of vines are negatively affected by soil salinity if enough salts accumulate in the root zone. As part of the current study, we estimated the remediating effects of rootstocks under salinity. For this reason, “superior seedless” vines were grafted onto three different rootstocks, such as SO_4_, 1103 Paulson, and own-root (“superior seedless” with their own-root). The experiment was conducted in the 2019 and 2020 seasons. This study examines the effects of different rootstocks on vine growth, yield, and quality using “superior seedless” vines grown in sandy soil with salinity. Four stages of berry development were examined (flowering, fruit set, veraison, and harvest time). At harvest, yield characteristics (clusters per vine and cluster weight) were also assessed. Each parameter of the growth season was influenced separately. The K^+^ and Na^+^ ratios were also significantly increased, as were the salinity symptoms index and bunch yield per vine and quality. Rootstock 1103 Paulson improved photosynthetic pigments, K^+^ accumulation, Na^+^ uptake, and cell membrane damage in “superior seedless” vines compared to other rootstocks, according to the study results. As determined in the arid regions of northwestern Egypt, the 1103 Paulson can mitigate salinity issues when planting “superior seedless” vines on sandy soil.

## 1. Introduction

Grapes are a high-income food crop with a high economic value, making them a valuable crop for producers. Mediterranean wine grapes are grown commercially in many different regions [1], and Egyptian agriculture has succeeded in increasing their vineyard area by 221,709 hectares over the past decade, yielding 1,626,259 tons of grapes [2]. Nevertheless, viticulture has made great strides. Despite all that has been accomplished, producers in these regions still face certain threats. One of the threats to viticulture in Egypt is soil salinity. ECe (electrical conductivity of the saturation extract) of 1.5 ds m^− 1^ is considered a lower tolerance to soil salinity in grapes, showing a decrease in growth with each new unit of ds m^−1^ increase [3].

This problem has forced the state, particularly on land adjacent to northern Mediterranean Sea coasts, to pay attention to many root assets to overcome it. As a result, some of the rootstocks used in this study have the following characteristics. The SO_4_ rootstock is a hybrid of *V. berlandieri* and *V. riparia*, and was developed by Oppenheim, Germany. SO_4_ is highly adaptable to wet, saline, and acidic soils [4]. One of the most notable features of rootstock 1103 Paulson is its high tolerance to drought and alkaline soil. It also thrives in soils with high calcium content [5].

Furthermore, rootstocks play several important roles in grape-growing countries. They are usually used in two important ways: first, to treat diseases that affect grape cultivation, like phylloxera and nematodes [6,7]. Second, to prevent these diseases from spreading. The second trend is to use assets to improve soil, increase plant resistance to salinity, lower soil pH, or poor drainage and drought [8,9,10,11,12]. The rootstock affects scion growth, yield, and bunch quality [13,14,15]. For the vine canopy (scion) to grow, rootstocks (roots) should absorb and transport most of the water, nutrients, and hormones required. As a result, the potential for vine tissue dehydration in response to abiotic stresses is different [16,17]. For optimal vine yield, the root system is required [18]. Water flow, hormone concentrations, nutrient acquisition and assimilation, and finally, the graft union anatomy have been suggested as explanations for the scion vigor conferred by rootstock [19,20].

The study estimates the importance of different rootstocks of the “superior seedless” vines grown in sandy soil with salinity on vine growth, yield, and fruit quality.

## 2. Results

### 2.1. Salinity Injury Index (SI-Index)

The salinity injury index (SI-index) for all rootstocks of the “superior seedless” vine is plotted as a function of the berry developmental stage (BDS). All rootstocks of the “superior seedless” vine are presented in Figure 1. The SI-index reveals a significant effect at *p* < 0.05, whereas the rootstocks were considered as a factor. Generally, from the reality of the results obtained, SI appeared with all rootstocks to different degrees throughout the BDS. Compared to the other rootstocks, the 1103 Paulson rootstock has slight symptoms of salinity on leaves. With vines grafted on the 1103 Paulson rootstock, we saw no sign of SI on leaves until the veraison stage, then a modest increase was observed until the end of the harvesting stage. 

The injury signs were discovered before the fruit set stage on vines grafted on SO_4_ and self grafted rootstocks and enlarged quickly up to the veraison phase, with more development until the end of the season. The self grafted rootstock, on the other hand, consistently produced the most damage signs throughout the berry development stages.

### 2.2. Chlorophyll Fluorescence Parameters

Table 1 shows the F_v_/F_m_ ratio, measured on a dark-adapted leaf. The effect of rootstocks was delineated as a function of vegetative growth stages. A significant interaction was found when vegetative growth stages and rootstocks were examined. First, we found a significantly decrease in the F_v_/F_m_ own-grafted rootstock compared to SO_4_ and 1103 Paulson rootstocks. We observed that all rootstocks initially presented a similar F_v_/F_m_ ratio reduction initially, but after the veraison stage, F_v_/F_m_ had directly decreased more rapidly. The reduction of the F_v_/F_m_ was due to the salinity relationship that occurred at the fruit set stage for all rootstocks. The “superior seedless” graft on the own-root rootstock, on the other hand, had a lower F_v_/F_m_ ratio than the other rootstocks. This difference was significant at around 0.65. F_m_ and F0 rates improved significantly in all rootstocks used in this study at the start of the growing season in the veraison stage. Concerning the impact of the rootstocks on F_0_ data, it became clear that the values increased irrespective of the type of rootstocks throughout the increasing BDS. The F_0_ values had changed because of rootstocks used. In a similar vein to F_m_, F_0_ initially (at the flowering stage), increased almost twice up to the veraison stage with all rootstocks. However, it gradually decreased, almost to the initial values at the harvesting stage. The 1103 Paulson rootstock had the highest F_0_ values compared to the SO_4_ rootstock. Nevertheless, the own-root rootstock showed a lower increase in F_0_ compared to other rootstocks, then gradually decreased, relatively close to the initial values.

### 2.3. Photosynthetic Pigment (Chlorophyll and Carotene Content)

Table 2 depicts the differences in overall chlorophyll pigment compounds (Chl a, Chl b, and Chl a + b) and carotene in all rootstocks as a feature of BDS. A significant interaction (*p* < 0.001) was obtained for a degree of vegetative growth of “Superior seedless” (Scion) increase because of rootstocks. The rootstocks afforded an increase in the Chls content in three Chls compounds throughout the BDS (four stages). In comparison, rootstocks had lower Chl a + b parameters initially, following an increase in all Chls parameters more rapidly throughout both seasons. The 1103 Paulson rootstock had the highest Chls content during the developmental stages of berries for Chl a, Chl b, and Chl a + b. The Chls content parameters increased with increasing BDS for all rootstocks. We also observed that the initial concentration of Chl a + b parameters differed due to the rootstocks and gradually increased over the growing season. All Chls parameters increased during the growing season. In the case of 1103 Paulson rootstock, there was high chlorophyll content during growth stages for all chlorophyll parameters. However, the other rootstocks presented the same trend but lost an increase in chlorophyll content parameters. The chlorophyll parameters increased rapidly during growth stages with the 1103 Paulson rootstocks, faster than on any other rootstock. However, other rootstocks presented the same trend, but chlorophyll content parameters reduced more rapidly during the growth season. Table 3 also shows the rest of the Chls compounds and other photosynthetic pigments such as the chlorophyll a/b ratio (Chla: b ratio), carotene (Car), and the ratio of Chl: Car plotted against the BDS for “superior seedless” vine grafted on three rootstocks. We noticed that the initial Chla: b ratio values were deferred among rootstocks. Also, the differences in Chls and Car between rootstocks are independent because of the rootstocks effect on “superior seedless” (scion). The 1103 Paulson rootstock provided the highest Chla: b ratio in the flowering stage compared to other rootstocks used. However, the own-root rootstock, showed almost the lowest Chla: b ratio during BDS. However, the leaf Car pigment content of the “superior seedless” vine grafted on the 1103 Paulson rootstock is the highest value compared to other rootstocks.

In general, we discovered that the Car content increased as BDSs increased, increasing up to the veraison stage and then decreasing until the harvesting stage. The own-root rootstocks showed almost stable content during BDS. The Chl: Car ratio, exhibited an increase in rates from the flowering up to the harvesting stage of BDS. After that, it increased continuously until the bunch harvest, which was observed with all rootstocks. Further, the 1103 Paulson rootstock had the highest values than those shown with other rootstocks throughout the growth stages.

### 2.4. Leaf Area, Shoot Carbohydrate, Proline, and Glycine Accumulation

The leaf area and the carbohydrate content were estimated as a power of the BDS of the grafted “superior seedless” vine on several rootstocks (Table 3). The variables presented a significant interaction when BDSs and rootstocks were considered. The 1103 Paulson rootstock positively affected the leaf area and shoot carbohydrate accumulation greatly under soil salinity compared to other rootstocks. This result was reproduced in the rate of leaf area and the increasing shoot carbohydrate content throughout the BDS. The rootstock 1103 Paulsen effect on the leaf area of “Superior seedless” (131.59 cm^2^) compared to the effect of the SO_4_ (116.73 cm^2^) and the own-root rootstock (103.79 cm^2^) at the harvesting stage. Another sign of the effect of the rootstock of 1103 Paulsen was accreted in the carbohydrate amount of the shoot at the harvest stage (33.55%) with growing the “superior seedless” vine on 1103 Paulson rootstock. However, the other rootstocks recorded SO_4_ (27.43%), and own-root (29.44%). proline and glycine levels, which were assessed during the four stages of berry development (Table 3). Generally, both variables increased independently because of rootstocks. However, the 1103 Paulson rootstock increased accumulation in both proline (0.39 mg g^−1^ FW) and glycine (2.20 µg g^−1^ FW) at the harvest stage compared to other rootstocks.

### 2.5. Malondialdehyde (MDA), and Electrolyte Leakage Percentage (EL%)

Table 4 presents the variation in MDA and Electrolyte Leakage (EL)% of the “superior seedless” vine (scion) grown on three rootstocks as a function of BDS. Further, it presents a significant interaction (*p* < 0.001) of all parameters plotted when BDS and rootstocks were considered factors. The MDA and EL% were also noticeable changes due to the different rootstock, compared to the “superior seedless” vines grafted on different rootstocks. We observed that the 1103 Paulson rootstock afforded lower MDA content and EL% compared to other rootstocks during BDS in both seasons (Table 4).

### 2.6. Na^+^, K^+^, and K^+^/Na^+^ Content of Leaves

Table 5 shows the changes in the ion percentage, leaf mineral content (K and Na ions) of the “superior seedless” vine (scion) grown on three rootstocks as a function of BDS. Further, it presents a significant interaction (*p* < 0.001) of all parameters plotted when BDS and rootstocks were considered factors. Rootstocks significantly impacted Na^+^ and K^+^ content and K^+^/Na^+^ ratio in “superior seedless” (scion) leaves, both under soil salinity stress. Soil salinity, Na^+^ increased while K^+^ and the K^+^/Na^+^ ratio decreased. The 1103 Paulson rootstock applied resulted in a decreased Na^+^ content and increase in K^+^ concentration and the K^+^/Na^+^ ratio up to the harvest stage.

### 2.7. Growth, Yield, and Bunch Quality

Table 6 and Table 7 present the growth, yield, and berry quality properties. The quality variables were significantly affected by vine rootstocks at 5%. Regarding the growth parameters, the 1103 Paulson rootstock provided more noticeable effects of internode length and thickness, wood pruning weight, and leaf area at harvest time compared to other rootstocks.

### 2.8. Data Correlation

For the studied physiological traits in response to different rootstocks of “superior seedless” vines during four levels of BDS (Table 8 and Figure 2), a variation of 59.5% was noted in PC1, and a variability of 18.2% was observed in PC2. The variables EL%, MDA, and Na^+^ percent were positively correlated with the SI- index. In contrast, all the other variables were negatively correlated, even if many were in an insignificant way, with the SI-index. Chlorophyll A and B concentrations and total chlorophyll content correlated positively with chlorophyll fluorescence determinations. These variables (SI-index, MDA, and Na^+^ percent) correlated negatively with the further variables. Chl B had an inverse relationship, including Chl A: B. The correlated positively with Chls: Caro and F_v_/F_m_, but negatively with the other variables. 

## 3. Discussion

Salinity is considered a serious global issue, affecting plant growth and productivity. Plants exposed to salinity first experience osmotic stress, then cell turgor-dependent activities, and finally vital physiological processes [21]. Numerous studies have demonstrated the possibility of abiotic stressors occurring in agricultural fields and their detrimental effect on plant productivity. Inappropriate agricultural techniques have degraded soil quality and fertility, which has led to a decrease in the amount of land available for agriculture worldwide [22,23]. It seems that the different responses of “superior seedless” rootstocks are due to different accumulations of Na and Cl which are indirectly involved in the toxicity of change in the natural balance of the vine [24]. Consequently, all rootstocks responded independently and showed different levels of salt injuries or symptoms (leaf burn, defoliation, and shoot necrosis). This may be due to the genetic difference between the rootstocks [25], which affects the scions (“superior seedless”). The outcomes from Table 2 and Table 7 pointed out the different responses of “superior seedless” rootstocks to soil salinity.

In We observed that the F_v_/F_m_ ratio was affected by rootstock effects during the seasons. Therefore, it appears that variations in the F_v_/F_m_ ratio as well as the changes in chlorophyll content or cell membrane damage could be used to monitor the BDS [26]. Furthermore, the Chl A: B ratio reacted differently due to the effects of rootstocks. [27,28]. Therefore, the differences in Chl and Car pigments content material may be associated with the changes in weather from warm to cold days with a common moisture deficit [29,30]. The increase in the Chl: Car ratio perhaps indicates that this is due to the increasing accumulation of the Car pigmen (Table 3). Basically, the carotenoid performs as a stabilizer of the light-center-harvesting protein of PS-I and PS-II. Additionally, it acts to guard the Chl pigment against the dangerous photo-unfavorable response through oxygen radical generation under stress [31]. Additionally, the Chl: Car ratio results observed became otherwise because of special preceding research on grapes, including [32]. However, the ratio that was observed barely decrease after the Chl: Car ratio was recorded in variety 3: 4 as observed and found by [33,34]. As described in the results, the behavior of the photosynthetic pigment during BDS appears to be independent of the rootstocks being used. Although the increase in leaf senescence was fast under salinity, it had an increase in pigments. It could be due to applying some biofertilizers such as humic acid, which can also increase the salt tolerance of vines and reduce soil salinization. [35].

From the results obtained, this study proved that the rootstock 1103 Paulson gives the “superior seedless” (scion) more resistance to soil salinity, which could be reflected in the efficiency of the photosynthesis process that is not related to the amount of chlorophyll [36]. The previous indicative variables gave significant results during the BDS. Furthermore, soil salinity negatively affects leaf area development [37], and the rootstocks differ in the extent of their response to the salinity of the soil genetically [38]. Lowered proline and glycine content was monitored with the own-root rootstocks. This directly resulted in growth losses and may be because of the effect of soil salinity and shortage of mineral absorption [39,40]. In addition, the rootstocks used showed different soil salinity behaviors concerning different carbohydrate content, possibly due to the toxic effect of both sodium and chloride ions on vines [40,41]. Also, the toxic effect affects the chlorophyll content [37]. Under the stress of soil salinity, the grapevines reduce the internal osmotic pressure of the cells of some inorganic elements such as potassium [40]. The vines also increase or maintain constant levels of proline and glycine [39], which reduces the toxic effect of sodium [41,42].

However, in vines growing under salinity stress, the oxidative reaction is caused by increasing reactive oxygen species in root tissue cells, leading to enhanced lipid peroxidation processes [43], a result of which is increased cell permeability afterward [44], as well as disrupting the metabolic equilibrium [45]. Salinity stress induces the creation of H_2_O_2_, which hydroperoxides the unsaturated fatty acids in bio-membranes and is produced as a byproduct of aldehyde breakdown, such as MDA [46,47,48]. All these factors result in cell disruption [49] and full photosynthesis loss [50], consistent with the observed EL% and MDA accumulation in this work (Table 4). Additionally, magnesium buildup is toxic to cells and causes various adverse effects, including protein breakdown [50].

In the present study, “superior seedless” rootstocks behaved differently during the absorption and distribution of mineral elements. Under salinity stress, the 1103 Paulson rootstock was better than the other rootstocks at accumulating K^+^ and minimizing Na^+^ (Table 5). It was observed that the increase in K^+^ and the decrease in Na^+^ changed independently according to the rootstock type in the study. These differences between the rootstocks may be mainly due to the increase in the rate of transpiration from vines growing under the stress of salinity, which leads to an increase in the osmotic pressure in the rhizosphere of the vine [51]. The reflection of this reduced the growth of vines due to reducing the negative osmotic pressure in the root zone, with an increase in the toxic effect of both sodium and chloride ions [52,53]. From this attitude, the main component is the vine absorption of potassium, but absorption is impeded under the influence of salinity stress [54]. Thus, enough potassium must be absorbed to meet the growing needs of vines and before the growth of clusters [55].

The current work could explain the 1103 Paulson rootstock by the enhanced uptake of macronutrients (NPK) that increased photosynthesis performance during growth stages [56]. Consequently, carbohydrate content increased at the end of the season [57]. Our results proved that the 1103 Paulson rootstock increased potassium uptake during growth and development, which increased vine yield more than other rootstocks (Table 5) and the internode length and thickness and leaf area (Table 3). However, the berry quality properties were significantly impacted by growing “superior seedless” (scion) on 1103 Paulson rootstock. It had significantly better bunch quality than the other rootstocks tested (Table 6 and Table 7). It might be perceived that the lowermost SSC: TA-ratio with 1103 Paulson rootstock could be genetically advantageous to increase the shelf life of bunches much more than other rootstocks. Our results were agreed with theses findings [58,59].

## 4. Materials and Methods

### 4.1. Vine and Field Experimental Setup

A commercial vineyard in the Nobaria area of Egypt (31.23° N, 29.96° E), was studied for two growth seasons (2019 and 2020). “Superior seedless” vines were 10 years old and were grafted on three rootstocks (SO_4_, and 1103 Paulson, and own-root). 

The following characteristics of rootstocks were used for “superior seedless”: The SO_4_ (*V. beriandiri* × *V. riparia*) is highly resistant to phylloxera and a medium for nematodes and calcareous soil. The 1103 Paulsen (*V. Berlandieri* × *V. rupestris*) is extremely vigorous, medium for the nematode, and adapted to calcareous soil [60,61]. The vines were grown in a 3 m × 3 m bed of sandy soil with drip irrigation system. Pruning level was performed on all vines at 70 bud vines^−1^ (7 cans × 10 buds can^−1^ each on four cardons), and the Y system trained all vines. Aside from that, all vines were subjected to the same famed management precautions, with three replicates of four vines in each. The treatment order was as follows: As a control, vines were grafted onto their rootstock, followed by SO_4_ and then 1103 Paulson rootstock. Soil and irrigation water were also analyzed to determine the growth stage of the vines [62,63,64] and is shown in Table 9.

### 4.2. Salt Injury Index (SS-Index)

Salinity injury symptoms are indicated or evaluated by looking at the necrosis spots of the leaf and leaf edges. Though subjective in reality, the suggestion appears to find application in most product harm evaluations. It is common for us to use an assessment of visual damage as a way to try to correlate the most effective techniques for following or understanding the evolution of salinity, such as chlorophyll measurements, EL measurements, and chlorophyll fluorescence measurements [24]. (1) a leaf with no necrotic tissues; (2) a blade with slight necrosis symptoms (the tip of the leaves); (3) a stem with moderate necrosis; (4) a stem with severe necrosis; and (5) a stem with very severe necrosis that results in dead tissue. At every stage of the development of the berry, the SS-Index was determined and recorded (BDS).

### 4.3. Chlorophyll Fluorescence (CF) and Photosynthetic Pigment Analysis

The measurement was conducted on the 7th leaf from the base of the shoot (at vine top) and was measured during the berry development stage using chlorophyll fluorescence (CF) measurements. Fluorimeter model Mini-PAM, (Win Control, Walz, Effeltrich, Germany), was used to obtain measurements during all four developmental stages. The F_v_/F_m_ ratio (percentage of shifting from minimum to maximum fluorescence, where F_v_ = F_m_ − F_0_) is the CF variable, along with F_0_ and F_m_ (the light-saturated yield of fluorescence), respectively. F_v_/F_m_ ratios between 0.75 and 0.78 indicate photosynthesis dysfunction due to a deficit in the electron transfer ability of photosystem II [65]. A spectrophotometer was used to determine the total amount of chlorophyll and carotene, the pigments responsible for photosynthesis [66]. Leaf sample (2.5 g) chlorophyll and carotene content were estimated using 5 mL of N, N-Dimethylformamide according to the method [67].

### 4.4. Leaf Area and Shoot Carbohydrate Accumulation

The leaf area measurement was determined on the complete leaf using a Sokkia Planix 7 digital planimeter. Ten leaves were selected from the vine and it represented cm^2^ [68]. The can carbohydrate accumulation was measured by applying the method of [69,70]. 

### 4.5. Leaf Proline and Glycine Content

Seventh leaf, proline quantification, and 100 mg leaf samples from a saline-stressed vine were homogenized in 3 percent sulfuric acid and centrifuged at 5000 rpm for 10 min. To this was added 20 mL of 6 M Phosphoric Acid and maintained at 40 °C for 24 h. Ninhydrin and glacial acetic acid were added to 2 mL plant extract, and the mixture was heated to 100 °C for 1 h. Then the reaction was stopped by cooling the solution. Toluene (4 mL) was added and aggressively mixed for a few seconds before measuring the optical density of the colored component at 520 nm. [71]. The leaf glycine betaine content was measured on the 7th leaf. Dry leaf powder (0.5 g) was shaken with 20 mL deionized water for 48 h at 25 °C. The samples were then filtered and frozen until analysis. 1:1 dilution of thawed extracts in 2 N sulfuric acid an aliquot (0.5 mL) was placed in ice water for 1 h. Then 0.2 mL of cold potassium iodide-iodine reagent was gently mixed with the vortex mixture. It was stored at 0–4 °C for 16 h. Samples were transferred to centrifuge tubes and centrifuged at 10,000 g for 15 min at 0 °C. 1 mL micropipette of the aspirated supernatant. As the solubility of periodate complexes in acid increased with temperature, the tubes were kept cold until the periodate complex was separated; 9 mL 1,2-dichloroethane (reagent grade) dissolved the periodate crystals. Strong vortex mixing was used to achieve complete solubility. The absorbance at 365 nm was measured after 2.0–2.5 h. The sample estimation procedure was used to prepare glycine betaine reference standards (50–200 g/mL) in 2 N sulfuric acid [72] concentrations were measured on the base shoot (mg g ^−1^ FW).

### 4.6. Leaf Mineral Content

During four stages of berry development, the mineral content of the seventh leaf on the base shoot was determined—amount of potassium [63]. The sodium content percentage was another important factor to consider [64]. As a percentage, all mineral content was presented.

### 4.7. Malondialdehyde (MDA) and Ion Leakage%

The TABRs test was used to measure lipid peroxidation accumulation. To 25 mL of alcohol, add metaphosphoric acid (5% *w*/*v*; HPO_3_) and butylated hydroxytoluene (BHT) (2%; C_15_H_24_O) and homogenize. We prepared a standard curve using 1,1,3,3-tetra ethoxy propane (C_7_H_16_O_4_; Sigma–Aldrich, St. Louis, MO, USA) that was comparable to malondialdehyde (MDA) at concentrations of 0.01–1.0 mg/L (TBARS) to estimate MDA accumulation [73].

The EL% of all samples was estimated at various stages. For three hours, 2 g of rachis was added to 10 mL of mannitol at a temperature of 6 moles per liter. The conductivity was read with a conductivity meter (M1). All cuvettes were boiled at 100 °C for one hour in a water bath to destroy the peel tissue. We then re-read the conductivity of all cuvettes to determine if there had been any total leakage (M2). The calculation of the relative ion leakage as a percentage [66].

### 4.8. Growth, Yield and Bunch Quality

Measurements were made of internode length (cm), internode thickness (cm), and leaf area (cm^2^). When the fruit was harvested, yield and berry properties such as the number of clusters per vine, average cluster weight (g), and yield per vine (kg) were measured, while wood pruning weight (kg vine^−1^) was determined during winter pruning. A refractometer was used to measure total soluble solid content (SSC%), total acid percentage (TA%), and the SSC: TA-ratio [74].

### 4.9. Statistical Analysis

The research was outlined as a completely randomized factorial with three factors: rootstocks (three rootstocks), seasons (two seasons: 2018 and 2019), and berry developmental phases (four phases) with three replicates per rootstock. The Pearson relationship matrix and principal component analysis were conducted for the studied physiological traits in response to different rootstocks of “superior seedless” vines during four levels of BDS. Tukey’s-HSD Test was applied using JMP Pro 16 software, with *p* < 0.05 taken as statistically significant (SAS Institute, Cary, NC, USA).

## 5. Conclusions

Based on these findings, 1103 Paulson in symbiosis with “superior seedless” rootstocks helped increase salinity stress tolerance, minimized the Na^+^ uptake, and increased the K^+^/Na^+^ ratio. Therefore, it can be concluded that farmers must do is determine the optimal choice of rootstock (1103 Paulson) that is compatible with the place of cultivation, especially with sandy soil under saline conditions.

## Figures and Tables

**Figure 1 plants-10-02215-f001:**
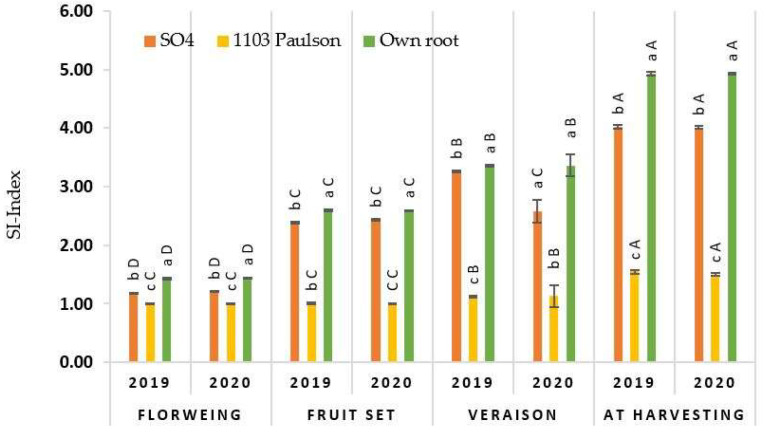
The impact of several rootstocks on ‘superior seedless’ vines during four berry developmental phases (flowering, fruit set, veraison, and harvesting stages) under soil conditions on the salinity injury index. The values represent the mean levels of expression in each treatment SE (*n* = 3). Tuckey’s HSD test (*p* < 0.05) used mean sorting within poles (uppercase letters) to identify significant differences between growing seasons and berry developmental phases (lowercase letters) to distinguish significant variations among Mg forms.

**Figure 2 plants-10-02215-f002:**
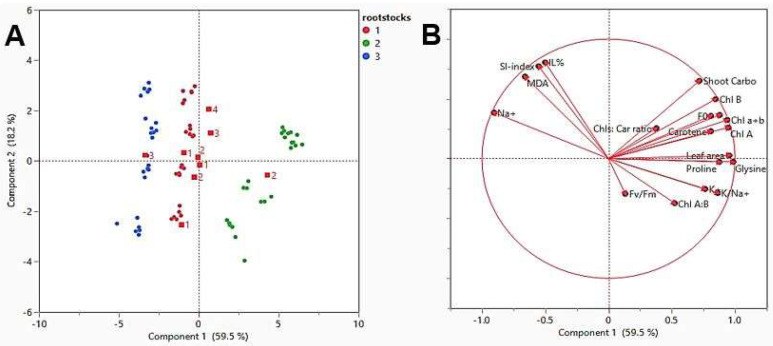
The principal component analysis (PCA) drafted with the participation of each variable on the two PCA axes (**A**) and all the metabolic aspects variables estimated in leaf over the growing season for the “superior seedless” vine grown in sandy soil and salt situations (**B**). Variable correlation using principal component analysis (PCA).

**Table 1 plants-10-02215-t001:** The effect of rootstocks (SO_4_, 1103 Paulson, and own root) on chlorophyll fluorescence variables (F_v_/F_m_ ratio, F_m_, and F_0_) was determined four times during two growth seasons (2018–2019) on various berry developmental phases (flowering, fruit set, veraison, and at harvest time) of “superior seedless” vines.

Variables	Rootstocks	Flowering		Fruit Set		Veraison		At Harvesting	
		2019	2020	2019	2020	2019	2020	2019	2020
F_v_/F_m_	SO_4_	0.806 ± 0.002 b A *	0.805 ± 0.000 b A	0.815 ± 0.002 b A	0.808 ± 0.000 b A	0.791 ± 0.004 b B	0.782 ± 0.001 b B	0.728 ± 0.001 b C	0.729 ± 0.000 b C
	1103 P	0.837 ± 0.004 a A	0.838 ± 0.000 a A	0.858 ± 0.000 a A	0.839 ± 0.000 a A	0.842 ± 0.000 a A	0.826 ± 0.001 a A	0.805 ± 0.002 a A	0.802 ± 0.001 a A
	Own root	0.798 ± 0.001 b A	0.799 ± 0.000 c A	0.803 ± 0.001 c A	0.798 ± 0.000 c A	0.785 ± 0.002 b B	0.779 ± 0.000 b B	0.628 ± 0.001 c C	0.634 ± 0.000 c C
F_m_	SO_4_	1885.00 ± 2.081 b D	1881.00 ± 0.577 b D	2115.33 ± 2.603 b C	2109.00 ± 0.577 b C	2326.66 ± 2.603 b A	2319.00 ± 0.577 b A	2274.33 ± 1.763 b B	2271.00 ± 1.000 b B
	1103 P	2136.00 ± 3.214 a D	2133.66 ± 1.201 a D	2344.33 ± 2.603 a C	2336.00 ± 1.527 a C	2564.33 ± 2.185 a A	2565.67 ± 2.603 a A	2483.00 ± 1.154 a B	2490.00 ± 0.577 a B
	Own root	1640.00 ± 1.527 c E	1649.67 ± 0.881 c D	1872.66 ± 2.185 c C	1865.33 ± 1.763 c C	1996.67 ± 1.527 c A	1992.33 ± 1.201 c A	1894.33 ± 2.403 c B	1893.33 ± 1.201 c B
F_0_	SO_4_	402.66 ± 1.111 b D	408.00 ± 0.577 b D	441.00 ± 0557 b C	443.00 ± 0.577 b C	514.33 ± 1.763 b A	517.66 ± 0.881 b A	476.00 ± 1.855 b B	477.00 ± 0.577 c B
	1103 P	421.00 ± 1.728 a D	423.00 ± 1.154 a D	536.66 ± 2.333 a C	536.00 ± 0.577 a C	881.33 ± 0.881 a A	882.00 ± 0.577 a A	765.66 ± 1.763 a B	770.00 ± 0.577 a B
	Own root	385.00 ± 1.081 c D	385.00 ± 0.577 c D	427.33 ± 1.855 c C	428.66 ± 0.881 c C	483.33 ± 2.027 c A	485.00 ± 0.577 c A	439.66 ± 1.855 c B	482.00 ± 1.154 b A

* The mean and standard error of the mean are used to represent the data. Tukey-HSD test at *p* < 0.05 for mean separation among columns (lowercase letters) and rows (uppercase letters). Data were obtained at various stages of berry growth.

**Table 2 plants-10-02215-t002:** The effect of rootstocks (SO_4_, 1103 Paulson, and own root) on photosynthetic pigments was assessed four times during two growth seasons (2018–2019) on various berry developmental phases (flowering, fruit set, veraison, and at harvest time) of “superior seedless” vines.

Variables	Rootstocks	Flowering		Fruit Set		Veraison		At Harvesting	
		2019	2020	2019	2020	2019	2020	2019	2020
Chl A (mg 100 g^−1^ FW)	SO_4_	1.57 ± 0.014 b D *	1.64 ± 0.008 ab C	1.68 ± 0.008 b C	1.68 ± 0.008 b C	1.80 ± 0.014 b AB	1.76 ± 0.005 b B	1.84 ± 0.005 b A	1.85 ± 0.008 b A
	1103 P	1.86 ± 0.008 a E	1.88 ± 0.011 a E	2.14 ± 0.020 a C	1.96 ± 0.008 a D	2.25 ± 0.029 a B	2.19 ± 0.008 a BC	2.35 ± 0.011 a A	2.35 ± 0.005 a A
	Own root	1.47 ± 0.012 c C	1.38 ± 0.118 c BC	1.58 ± 0.014 c ABC	1.56 ± 0.005 c ABC	1.66 ± 0.005 c AB	1.63 ± 0.017 c AB	1.73 ± 0.008 c A	1.71 ± 0.005 c A
Chl B (mg 100 g^−1^ FW)	SO_4_	0.61 ± 0.003 b F	0.63 ± 0.005 b F	0.66 ± 0.005 b E	0.67 ± 0.008 b DE	0.70 ± 0.005 b CD	0.71 ± 0.005 b BC	0.73 ± 0.005 b AB	0.75 ± 0.005 b A
	1103 P	0.67 ± 0.008 a D	0.67 ± 0.005 a D	0.72 ± 0.005 a C	0.75 ± 0.005 a C	0.84 ± 0.008 a AB	0.86 ± 0.008 a A	0.82 ± 0.005 a B	0.82 ± 0.003 a B
	Own root	0.54 ± 0.005 c D	0.53 ± 0.005 c D	0.61 ± 0.005 c C	0.66 ± 0.005 c AB	0.64 ± 0.005 c B	0.66 ± 0.006 c AB	0.66 ± 0.005 c AB	0.68 ± 0.005 c A
Chl A+B (mg 100 g^−1^ FW)	SO_4_	2.18 ± 0.017 b F	2.27 ± 0.008 a E	2.34 ± 0.013 b D	2.36 ± 0.017 b D	2.50 ± 0.018 b BC	2.47 ± 0.001 b C	2.57 ± 0.011 b AB	2.60 ± 0.012 b A
	1103 P	2.54 ± 0.011 a E	2.55 ± 0.015 a E	2.86 ± 0.026 a C	2.71 ± 0.008 a D	3.09 ± 0.037 a AB	3.06 ± 0.017 a B	3.17 ± 0.017 a A	3.17 ± 0.008 a A
	Own root	2.01 ± 0.012 c BC	1.91 ± 0.113 b C	2.19 ± 0.008 c AB	2.22 ± 0.011 c A	2.31 ± 0.005 c A	2.29 ± 0.012 c A	2.39 ± 0.013 c A	2.39 ± 0.010 c A
Chl A:B ratio	SO_4_	2.56 ± 0.012 b ABC	2.61 ± 0.031 a ABC	2.55 ± 0.018 b A	2.48 ± 0.021 b BC	2.58 ± 0.017 b AB	2.48 ± 0.028 a BC	2.52 ± 0.011 c ABC	2.47 ± 0.020 b C
	1103 P	2.77 ± 0.040 a B	2.80 ± 0.024 a B	2.98 ± 0.006 a A	2.62 ± 0.028 a CD	2.67 ± 0.011 a C	2.53 ± 0.015 a D	2.87 ± 0.005 a B	2.84 ± 0.005 a B
	Own root	2.73 ± 0.041 a A	2.62 ± 0.245 a A	2.59 ± 0.046 b A	2.36 ± 0.011 c A	2.55 ± 0.027 b A	2.47 ± 0.046 a A	2.62 ± 0.016 c A	2.51 ± 0.018 b A
Caroten (mg 100 g^−1^ FW)	SO_4_	2.37 ± 0.012 b C	2.47 ± 0.065 a BC	2.46 ± 0.005 b BC	2.46 ± 0.005 b BC	2.63 ± 0.012 b A	2.66 ± 0.005 b A	2.51 ± 0.005 b B	2.50 ± 0.005 b B
	1103 P	2.58 ± 0.014 a D	2.57 ± 0.005 a D	2.91 ± 0.017 a C	2.91 ± 0.008 a C	3.05 ± 0.029 a B	3.05 ± 0.005 a B	2.91 ± 0.008 a C	3.93 ± 0.006 a A
	Own root	2.25 ± 0.020 c F	2.23 ± 0.014 b F	2.37 ± 0.012 c E	2.40 ± 0.005 c DE	2.53 ± 0.012 c AB	2.58 ± 0.008 c A	2.44 ± 0.008 c CD	2.49 ± 0.005 b BC
Chls: Caro ratio	SO_4_	0.92 ± 0.008 b B	0.92 ± 0.026 a B	0.95 ± 0.006 b B	0.96 ± 0.005 a B	0.94 ± 0.003 b B	0.93 ± 0.002 b B	1.02 ± 0.003 b A	1.04 ± 0.006 a A
	1103 P	0.98 ± 0.006 a C	0.99 ± 0.006 a BC	0.98 ± 0.003 a C	0.93 ± 0.001 b D	1.01 ± 0.003 a B	1.00 ± 0.006 a BC	1.08 ± 0.008 a A	0.80 ± 0.003 c E
	Own root	0.89 ± 0.008 b AB	0.85 ± 0.053 a B	0.92 ± 0.003 c AB	0.92 ± 0.006 b AB	0.91 ± 0.003 c AB	0.88 ± 0.008 c AB	0.98 ± 0.005 c A	0.96 ± 0.005 b A

* The mean and standard error of the mean are used to represent the data. Tukey-HSD test at *p* < 0.05 for mean separation among columns (lowercase letters) and rows (uppercase letters). Data were obtained at various stages of berry growth. Chl A: Chlorophyll A, Chl B: Chlorophyll B, Chl A+B: Total chlorophyll A+B, and the ratio between chlorophyll A and B, Car; Carotene, and Chls. Car ratio: The ratio between total chlorophyll and carotene content.

**Table 3 plants-10-02215-t003:** The effect of rootstocks (SO_4_, 1103 Paulson, and own root) on proline, glycine, shoot carbohydrate percentage, and leaf area (cm^2^) were assessed four times during two growth seasons (2018–2019) on various berry developmental phases (flowering, fruit set, veraison, and at harvest time) of “superior seedless” vines.

Variables	Rootstocks	Flowering		Fruit Set		Veraison		At Harvesting	
		2019	2020	2019	2020	2019	2020	2019	2020
Proline%	SO_4_	0.33 ± 0.048 a AB *	0.37 ± 0.005 a A	0.31 ± 0.005 b B	0.31 ± 0.005 b B	0.32 ± 0.005 b AB	0.33 ± 0.005 b AB	0.31 ± 0.005 b B	0.32 ± 0.005 b AB
	1103 P	0.33 ± 0.003 a D	0.35 ± 0.003 a CD	0.36 ± 0.005 a CD	0.36 ± 0.005 a CD	0.37 ± 0.005 a BC	0.39 ± 0.005 a AB	0.39 ± 0.008 a AB	0.40 ± 0.004 a A
	Own root	0.28 ± 0.005 a B	0.28 ± 0.003 b AB	0.29 ± 0.005 b AB	0.30 ± 0.005 a AB	0.30 ± 0.003 b AB	0.30 ± 0.005 c A	0.29 ± 0.005 b AB	0.31 ± 0.005 b A
Glycine%	SO_4_	1.47 ± 0.017 b E	1.48 ± 0.011 b E	1.59 ± 0.005 b CD	1.61 ± 0.003 b BC	1.65 ± 0.005 b AB	1.66 ± 0.008 b A	1.56 ± 0.005 b D	1.58 ± 0.005 b CD
	1103 P	1.96 ± 0.012 a E	1.97 ± 0.005 a E	2.14 ± 0.020 a CD	2.13 ± 0.003 a D	2.39 ± 0.017 a A	2.40 ± 0.005 a A	2.20 ± 0.011 a BC	2.23 ± 0.005 a B
	Own root	1.24 ± 0.008 c C	1.24 ± 0.005 c C	1.27 ± 0.015 c BC	1.27 ± 0.008 c BC	1.34 ± 0.005 c A	1.35 ± 0.005 c A	1.29 ± 0.005 c B	1.30 ± 0.005 c B
Shoot carboh. %	SO_4_	19.56 ± 0.594 b E	19.45 ± 0.583 b E	23.17 ± 0.551 b D	24.24 ± 0.586 b CD	26.36 ± 0.591 b C	26.43 ± 0.557 b C	29.44 ± 0.568 b B	32.85 ± 0.577 a A
	1103 P	22.84 ± 0.565 a C	22.74 ± 0.580 a C	25.75 ± 0.589 a B	27.44 ± 0.591 a B	31.84 ± 0.560 a A	32.22 ± 0.671 a A	33.55 ± 0.571 a A	34.36 ± 0.591 a A
	Own root	17.66 ± 0.594 b C	16.67 ± 0.571 c C	20.64 ± 0.591 c B	21.14 ± 0.597 c B	22.94 ± 0.589 c AB	25.33 ± 0.586 c A	24.64 ± 0.560 c A	25.64 ± 0.588 b A
Leaf area(Cm^2^)	SO_4_	108.12 ± 0.571 b E	109.92 ± 0.328 b DE	111.58 ± 0.864 b CD	112.35 ± 0.586 b CD	114.36 ± 0.574 b BC	116.67 ± 0.600 b AB	116.73 ± 0.568 b AB	117.37 ± 0.583 b A
	1103 P	127.62 ± 0.439 a D	128.01 ± 0.678 a D	129.31 ± 0.877 a CD	131.74 ± 0.594 a ABC	130.46 ± 0.586 a BC	133.67 ± 0.577 a AB	131.59 ± 0.562 a BC	134.73 ± 0.583 a A
	Own root	96.88 ± 0.548 c D	96.22 ± 0.887 c D	97.82 ± 0.583 c CD	97.31 ± 0.868 c D	102.69 ± 0.910 c AB	101.14 ± 0.571 c BC	103.79 ± 0897 c AB	105.27 ± 0.585 c A

* The mean and standard error of the mean are used to represent the data. Tukey-HSD test at *p* < 0.05 for mean separation among columns (lowercase letters) and rows (uppercase letters). Data were obtained at various stages of berry growth.

**Table 4 plants-10-02215-t004:** The effect of rootstocks (SO_4_, 1103 Paulson, and own root) on malondialdehyde (MDA), and electrolyte leakage percentage were assessed four times during two growth seasons (2018–2019) on various berry developmental phases (flowering, fruit set, veraison, and at harvest time) of “superior seedless” vines.

Variables	Rootstocks	Flowering		Fruit Set		Veraison		At Harvesting	
		2019	2020	2019	2020	2019	2020	2019	2020
MDA (nm 100 g^−1^ FW)	SO_4_	0.17 ± 0.005 b D *	0.18 ± 0.008 a CD	0.20 ± 0.005 a BCD	0.21 ± 0.005 a BC	0.23 ± 0.005 b AB	0.23 ± 0.005 b AB	0.26 ± 0.006 b A	0.26 ± 0.005 b A
	1103 P	0.14 ± 0.005 c AB	0.13 ± 0.006 b B	0.15 ± 0.008 b AB	0.14 ± 0.005 b AB	0.17 ± 0.005 c A	0.15 ± 0.005 c AB	0.17 ± 0.006 c A	0.16 ± 0.005 c AB
	Own root	0.20 ± 0.005 a C	0.20 ± 0.006 a C	0.22 ± 0.005 a C	0.22 ± 0.005 a C	0.26 ± 0.004 a B	0.26 ± 0.003 a B	0.31 ± 0.008 a A	0.33 ± 0.008 a A
EL%	SO_4_	10.76 ± 0.574 a E	11.25 ± 0.580 a 3	14.84 ± 0.574 a CD	12.34 ± 0.577 b DE	17.64 ± 0.580 b BC	19.65 ± 0.568 b AB	21.34 ± 0.565 b A	22.31 ± 0.568 b A
	1103 P	7.17 ± 0.598 b C	7.15 ± 0.571 b C	11.41 ± 0.859 b AB	9.44 ± 0.583 c BC	12.94 ± 0.580 c A	11.46 ± 0.583 c AB	13.23 ± 0.571 c A	13.54 ± 0.580 c A
	Own root	12.34 ± 0.580 a D	12.49 ± 0.588 a D	18.12 ± 0.864 a C	17.56 ± 0.588 a C	22.54 ± 0.589 a B	23.84 ± 0.597 a B	27.14 ± 0.831 a A	27.95 ± 0.583 a A

* The mean and standard error of the mean are used to represent the data. Tukey-HSD test at *p* < 0.05 for mean separation among columns (lowercase letters) and rows (uppercase letters). Data were obtained at various stages of berry growth. MDA: Malondialdehyde (MDA; ηM g^−1^ FW), EL%: Electrolyte leakage percentage.

**Table 5 plants-10-02215-t005:** The effect of rootstocks (SO_4_, 1103 Paulson, and own root) on potassium content percent (K%), sodium content (Na%), and the K/Na ratio were assessed four times during two growth seasons (2018–2019) on various berry developmental phases (flowering, fruit set, veraison, and at harvest time) of “superior seedless” vines.

Variables	Rootstocks	Flowering		Fruit set		Veraison		At harvesting	
		2019	2020	2019	2020	2019	2020	2019	2020
K%	SO_4_	1.62 ± 0.038 b F *	1.73 ± 0.015 b BC	1.67 ± 0.008 b DE	1.74 ± 0.012 b ABC	1.70 ± 0.014 b CD	1.76 ± 0.005 b AB	1.64 ± 0.005 b EF	1.78 ± 0.005 b A
	1103 P	1.80 ± 0.029 a D	1.83 ± 0.012 a CD	1.80 ± 0.015 a D	1.87 ± 0.005 a AB	1.86 ± 0.011 a BC	1.89 ± 0.005 a AB	1.81 ± 0.026 a D	1.90 ± 0.005 a A
	Own root	1.53 ± 0.020 c C	1.45 ± 0.015 c D	1.59 ± 0.005 c AB	1.56 ± 0.008 c ABC	1.60 ± 0.008 c A	1.59 ± 0.005 c AB	1.54 ± 0.014 c BC	1.56 ± 0.004 cABC
Na%	SO_4_	0.85 ± 0.012 b D	0.89 ± 0.011 b CD	0.93 ± 0.012 b C	0.91 ± 0.005 b C	1.09 ± 0.023 b B	1.07 ± 0.005 b B	1.16 ± 0.008 b A	1.16 ± 0.005 b A
	1103 P	0.28 ± 0.005 c CD	0.27 ± 0.005 c D	0.30 ± 0.005 c BCD	0.30 ± 0.008 c BC	0.32 ± 0.005 c AB	0.33 ± 0.005 c AB	0.33 ± 0.008 c A	0.35 ± 0.005 c A
	Own root	1.22 ± 0.005 a D	1.24 ± 0.005 a D	1.39 ± 0.018 a C	1.36 ± 0.005 a C	1.47 ± 0.020 a B	1.41 ± 0.005 a C	1.62 ± 0.011 a A	1.65 ± 0.005 a A
K/Na ratio	SO_4_	1.89 ± 0.051 b A	1.94 ± 0.013 b B	1.78 ± 0.021 b A	1.91 ± 0.008 b A	1.56 ± 0.034 b CD	1.64 ± 0.012 b C	1.41 ± 0.008 b E	1.53 ± 0.014 b D
	1103 P	6.43 ± 0.010 a AB	6.79 ± 0.101 a A	6.00 ± 0.026 a BCD	6.17 ± 0.185 a ABC	5.82 ± 0.187 a BCD	5.73 ± 0.083 a CD	5.38 ± 0.343 a D	5.43 ± 0.099 a D
	Own root	1.25 ± 0.020 c A	1.16 ± 0.017 c B	1.13 ± 0.008 c B	1.15 ± 0.008 c B	1.09 ± 0.006 c C	1.13 ± 0.001 c BC	0.93 ± 0.012 c D	0.94 ± 0.008 c D

* The mean and standard error of the mean are used to represent the data. Tukey-HSD test at *p* < 0.05 for mean separation among columns (lowercase letters) and rows (uppercase letters). Data were obtained at various stages of berry growth. K: Potassium content percentage, Na: Sodium content percentage, K/Na: The ratio between potassium and sodium.

**Table 6 plants-10-02215-t006:** The different between rootstocks of “Superior seedless” vine on growth and properties of selected bunches at harvest time.

Rootstocks	Internode Length (cm)	Internode Thickness (cm)	Leaf Area (cm^2^)	Bunch Weight (gm)	Bunch Number Vine^−1^	Yield Vine^−1^ (Kg)	Wood Pruning Weight (Kg Vine^−1^)
SO_4_	9.09 ± 0.011 ^ab^	1.29 ± 0.008 ^b^	121.36 ± 0.574 ^b^	321.15 ± 0.100 ^b^	19.33 ± 0.881 ^b^	6.21 ± 0.324 ^b^	15.88 ± 0.012 ^b^
1103 Paulson	10.13 ± 0.580 ^a^	1.55 ± 0.011 ^a^	136.22 ± 2.612 ^a^	424.68 ± 0.115 ^a^	26.01 ± 0.577 ^a^	11.04 ± 0.289 ^a^	17.39 ± 0.291 ^a^
Own root	7.78 ± 0.015 ^b^	0.86 ± 0.012 ^c^	113.56 ± 0.586 ^c^	301.77 ± 0.164 ^c^	16.00 ± 0.574 ^c^	4.83 ± 0.202 ^c^	15.05 ± 0.037 ^c^

The main data of both seasons were analyzed using a one-way ANOVA (complete block randomised design) of “Superior seedless” vines. Each value represents the mean and standard error (*n* = 3) of three replicates. The superscript letters differ (*p* < 0.05) and represent the significance between the main treatments using Tukey’s-HSD Test.

**Table 7 plants-10-02215-t007:** The different rootstocks of “superior seedless” vine on growth and properties of selected bunches at harvest time.

Rootstocks	SSC%	TA%	SSC:TA Ratio
SO_4_	16.17 ± 0.014 ^b^	0.760 ± 0.001 ^b^	20.99 ± 0.013 ^b^
1103 Paulson	14.28 ± 0.014 ^c^	0.805 ± 0.002 ^a^	18.77 ± 0.060 ^c^
Own root	16.89 ± 0.012 ^a^	0.694 ± 0.001 ^c^	23.29 ± 0.020 ^a^

The main data of both seasons were analyzed using a one-way ANOVA (complete block randomised design) of “Superior seedless” vines. Each value represents the mean and standard error (*n* = 3) of three replicates. The superscript letters differ (*p* < 0.05) and represent the significance between the main treatments using Tukey’s-HSD Test.

**Table 8 plants-10-02215-t008:** Shows the Pearson correlation matrix for the parameters of “superior seedless” vines grown on three rootstocks.

Variables	Chl A	Chl B	Chl a + b	Chl A: B	SI-index	Carotene	Chls: Car Ratio	Proline	Shoot Carbo	MDA	EL%	F_v_/F_m_	F_m_	F0	Glycine	Leaf Area	K^+^	Na^+^	K^+^/Na^+^
Chl A	*1.0000																		
Chl B	0.9064	1.0000																	
Chl a+b	0.9946	0.9453	1.0000																
Chl A: B	0.5334	0.1305	0.4439	1.0000															
SI-index	−0.2962	−0.1084	−0.2552	−0.4673	1.0000														
Carotene	0.8506	0.7905	0.8507	0.3878	−0.3108	1.0000													
Chls: Car ratio	0.4353	0.4386	0.4439	0.1950	0.1017	−0.0869	1.0000												
Proline	0.8047	0.7368	0.8021	0.3990	−0.5428	0.7566	0.2117	1.0000											
Shoot Carbo	0.8382	0.9315	0.8758	0.1035	0.1071	0.7382	0.4145	0.5873	1.0000										
MDA	−0.4190	−0.2517	−0.3852	−0.4694	0.9471	−0.4033	−0.0003	−0.6027	−0.0478	1.0000									
EL%	−0.2486	−0.0487	−0.2038	−0.4835	0.9128	−0.2055	−0.0036	−0.4523	0.1519	0.9194	1.0000								
F_v_/F_m_	0.0400	0.0044	0.0319	0.0858	−0.1774	−0.0026	0.0659	0.0574	−0.0382	−0.2246	−0.2240	1.0000							
F_m_	0.8972	0.9382	0.9230	0.2352	−0.2110	0.7684	0.4303	0.7259	0.8758	−0.3710	−0.1767	0.0257	1.0000						
F0	0.8466	0.8815	0.8700	0.2209	−0.2657	0.7923	0.2564	0.7045	0.7894	−0.3143	−0.1303	−0.0647	0.8072	1.0000					
Glycine	0.9098	0.8309	0.9064	0.4742	−0.5731	0.7843	0.3523	0.8420	0.6852	−0.6844	−0.5335	0.1282	0.8812	0.7956	1.0000				
Leaf area	0.8960	0.8213	0.8933	0.4609	−0.4696	0.7601	0.3937	0.8371	0.7162	−0.6160	−0.4801	0.1505	0.8857	0.6864	0.9612	1.0000			
K^+^	0.6498	0.4959	0.6234	0.5067	−0.5791	0.4190	0.4240	0.6313	0.3756	−0.5848	−0.4905	0.2292	0.5315	0.5213	0.7151	0.6521	1.0000		
Na^+^	−0.7384	−0.5839	−0.7134	−0.5589	0.7939	−0.6265	−0.2588	−0.7985	−0.4113	0.8808	0.7912	−0.2065	−0.6815	−0.5538	−0.9193	−0.8943	−0.7421	1.0000	
K^+^/Na^+^	0.7558	0.5601	0.7210	0.6359	−0.6286	0.5338	0.4178	0.6875	0.4142	−0.6854	−0.5879	0.2848	0.6105	0.5594	0.8423	0.7924	0.9210	−0.8643	1.0000

^*^ Values represent average values per season, rootstocks, and berry developmental phases. Chl A—Chlorophyll A content; Chl B—Chlorophyll B content; Chl a + b—Total chlorophyll content; Chl A: B—The ratio between chlorophyll A and B; Car—Carotene content; Chls: Car—The ratio between total chlorophyll and carotene; Proline; Shoot Carbo—Shoot carbohydrate content; EL%—Ion leakage percentage; MDA—Malondialdehyde accumulation; F_v_/F_m_—Chl fluorescence ratios; F_m_— Maximum Chl fluorescence in the light-adapted state; F0—ground fluorescence; Glycine; Leaf area (cm^2^); K^+^—Potassium content; Na^+^—Sodium content; K^+^/Na^+^—The ratio between potassium and sodium

**Table 9 plants-10-02215-t009:** Physical and chemical proprieties of the experimental soil and chemical study of drip irrigation.

Physical Properties						Soluble Anions (meq L^−1^)				Soluble Cations (meq L^−1^)				
Sand%	Clay%	Silt%	Texture	EC dsm^−1^	pH	HCO_3_^−^	Cl^−^	SO_4_		Na^+^	K^+^	Mg^++^	Ca^++^	SAR
86.7	7.7	11.93	Sandy	4.8	7.89	3.00	15.20	13.11		26	2.99	3.77	11.01	9.59
			Cations (meq L^−1^)											
pH	EC (dS m^−1^) 0.86		CO_3_^−^	HCO_3_^−^	Cl^−^	SO_4_^−^	Ca^++^	Mg^++^	Na^+^	K^+^				
7.28	563 ppm		0.21	2.64	0.92	1.21	1.78	0.73	25	0.17				

## Data Availability

Relevant data applicable to this research are within the paper.

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
