# Peer review of "Growth, Yield, and Bunch Quality of “Superior Seedless” Vines Grown on Different Rootstocks Change in Response to Salt Stress"

_plants, 2021, doi:10.3390/plants10102215_

Round 1

Reviewer 1 Report

This is a nice paper that reported results on the "Growth, yield, and bunch quality of 'Superior seedless' vines grown on different rootstocks change in response to soil salt stress". Although the data presented were quite comprehensive the manuscript itself needs major editing and reformatting before it is suitable for publication.

Here are my major comments:

  • English grammar and punctation need to be checked throughout the entire paper. There are errors even in the abstract, were improper punctation was used.
  • 'Superior seedless' need to be written in a consistent way throughout the manuscript: sometimes is 'Superior seedless', some others is "Superior seedless".
  • SO4 and the other 3 rootstocks need to be described in the material and methods section. Very few information has been given about the site, growing conditions, genotypes. I strongly suggest to add a paragraph with specific characteristics of the growing region, soil type, genotypes, weather etc etc
  • All the chemical ions need to be reported using subscripts (i.e., Na+)

I believe this paper has a good potential and the date presented are somewhat novel but the manuscript is not well prepared and organized. It looks messy and it seems that the Authors did not double check everything before submitting. I suggest to rework on the overall paper, make sure everything is well described and presented and then resubmit.

Author Response

  • English grammar and punctation need to be checked throughout the entire paper. There are errors even in the abstract, were improper punctation was used.

All manuscript parts were checked

  • 'Superior seedless' need to be written in a consistent way throughout the manuscript: sometimes is 'Superior seedless', some others is "Superior seedless".

The 'Superior Seedless' is consistent in the whole MS

  • SO4 and the other 3 rootstocks need to be described in the material and methods section. Very few information has been given about the site, growing conditions, genotypes. I strongly suggest to add a paragraph with specific characteristics of the growing region, soil type, genotypes, weather etc etc

All details of rootstocks were added in a paragraph in part 2.1 of M&M

  • All the chemical ions need to be reported using subscripts (i.., Na+

All checked in subscripts for ions

Reviewer 2 Report

The work requires extensive revision.

Starting with Materials and Methods because those for the quality of a job must be clear.

Before continuing the review of the work, I ask the authors to respond to these requests for clarification regarding M&M. 

The questions are in the attached file

Author Response

Response to revweirs

Review Report Form (Reviewer 2)

The work requires extensive revision. Starting with Materials and Methods, because those for the quality of a job must be clear and I find it very wrong that Plants has decided to put M&M at the end of a paper. M&M should be read first and if done badly you do not continue reading the entire work: The questions are in the attached file

  • M&M are often superficial and deficient I agree, but sometimes we need to identify some parameters to be clear such as chlorophyll fluorescence parameters
  • The vine plants subjected to the checks were very few (3x4 = 12 for each thesis)

This is what was available in that area, especially for this type of grafting and rootstocks

  • At lines 368-369, it non reported the trained system,

Was added

  • At line 370 was reported “Pruning was done on all vines at 70 bud vines-370 1 (7 cans X 10 buds can-1 each). What type of pruning?

Was added

  • Al line 379 the title is: “Salt symptoms index (SS-index)” but at line 388 is reported SI index and non SS-index (Why?)

Changes to SS-Index according to Salt Symptoms Index

  • At lines 390-399 were reported “Chlorophyll fluorescence and photosynthetic pigment analysis” Regarding the leaf sampling: was reitten in lines 441-442

Regarding foliar sampling for fluorescence measurements and chlorophyll analyzes:

o which leaves were sampled?

               7th leaf from the base of the shoot

o What position on the shoot?

Top of vines

Were they leaves exposed to light?

It misses typing to used dark-adapted

o Were they leaves showing SS (SI) symptoms?

Of course, its present low value in Fv/Fm (Figure 1)

 o No detail was reported about the determination of Chlorophyll and carotene.

             Both were added in lines 441-442

  • If the reference method [63] is used it is important to report it clearly.
  • Even more generic and unclear is what is reported in lines 401-406: o Was reported: “To measure cane leaf area and carbohydrate accumulation, the Sokkia Planix 7 digital planimeter planometer”.

A digital planimeter was used to determinate carbohydrate accumulatio?

Mistakes, it was arranged

Are you sure? o

At lines 404 was reported “Leaf Proline and glycine content” Is a title or sentence? More details about the method of analysis are required.

Both were added in detials

References [65] and [66] refer to really very old works.

Are there not more recent works that have addressed and updated these methods of analysis?

Update

  • At line 412 was reported Malondialdehyde (MDA) and ion leakage% o For the first time the procedure performed is reported in detail, fine. I will try to apply it to evaluate its effectiveness.

Thank you

Round 2

Reviewer 1 Report

The Authors addressed all of my comments. 

Author Response

Thank you

Our Refgards

Lo'ay, A. A.

Reviewer 2 Report

The comments are numerous and are reported in the attached file.
Thanks

Author Response

PART 2° for round 2 rewire 2

I thank the authors for the quick answer to my questions and the timely integration of M&M as requested. Now, however, many other, much more substantial, corrections must be made to the rest of the draft.

Thank you

Introduction

  • Line 24 reports “version”, but it is correct to report “verasion”. Beware of similar errors in the rest of the text, the autocorrect makes these kinds of mistakes!

Corrected in whole manuscript

  • I repeat that what is reported on lines 47-48 is incorrect, SO4 [Selection Oppenheim 4 (SO 4)] was not selected University of California, Davis. but at Oppenheim (Germany).
    • https://fps.ucdavis.edu/fgrdetails.cfm?varietyid=1116
    • https://www.vdp.de/en/die-winzer/rheinhessen/staatliche-weinbaudomaene-oppenheim-vdp
    • https://www.dlr.rlp.de/SWG-Oppenheim-englisch

Corrected in line 48-49 

Results

  • In line 66 was reported: 1. Salinity injury index (SI-index), but at line 383 (in M&M) we speak of Salt Symptoms Index (SS-Index). I had asked for this confusion to be cleared up in the previous review as well

Corrected sorry for miss that comment before. It was in lines 67 – 70  - 88  - 395 - 397

  • Line 68 reads: berry developmental stage (BDS).I propose that BDS is, from this point on, always used in place of the phrase 'berry developmental-stage

I agree with you thank you changed

  • Line 74 reports “version”, but it is correct to report “verasion”. Beware of similar errors in the rest of the text, the autocorrect makes these kinds of mistakes!

Corrected

  • The sentence on lines 96-97 needs to be changed. Fv / Fm is measured on leaves, not on rootstocks, therefore: “Table 1 shows the Fv / Fm ratio, measured on a dark-adapted leaf. The effect of rootstocks was delineated as a function of the vegetative growth phases.”

Corrected

  • If Fv/Fm was measured each time on the 7th leaf from the base of the shoot, as reported on M&M, the value is expected to decrease especially after the verasion time. It is a normal leaf senescence affect

Yes, also it is more responses to measure the effect of salinity to distinguish the differences between rootstocks impacts than the oldest and newest one

  • Line 101 reports “version”, but it is correct to report “verasion”. Beware of similar errors in the rest of the text, the autocorrect makes these kinds of mistakes!

Corrected in the whole manuscript

  • In lines 101-102 was reported: The reduction of the Fv/Fm is due to the salinity relationship that occurred at the fruit set stage for all rootstocks Given that this is possibly a consideration to put in the “Discussion” chapter. What is meant by this sentence? Does it mean that from fruit set onwards the damage caused by salinity has increased YES? First, a relationship must be found between damage on the leaves caused by salinity and the reduction of Fv / Fm. This could be achieved by taking the salinity damage measurements and fluorescence measurements on the same leaves. But in M&M there is no hint on which leaves the salinity damage estimate was made (SS or SI index).
  • In Table 1 (and in all other table) the values of SE appear strange. How was SE calculated. It is the standard error of the means, so each mean should have its own SE. Why SE it the same. Why is SE the same for the different averages of each individual rootstock at the same sampling point? There is clearly an error in the SE calculation. Please check all SE calculations in all tables.

All SE of our data corrected

  • In line 104 (and in all other case) F0 must be replaced with F0

Replaced

  • Line 106 reports “version”, but it is correct to report “verasion”. Beware of similar errors in the rest of the text, the autocorrect makes these kinds of mistakes!

Corrected in the whole manuscript

  • Why, at line 108, they say, “irrespective of the type of rootstocks.”? Indeed, the data show that rootstock influences the variation of F0 during the growing season. However, the data of Fm, F0 and Fv/Fm appear strange, generally as the season progresses Fm decreases, F0 increases and Fv/Fm decreases. In particular, numerous bibliography has shown that F0 increases in conditions of biotic stress (temperatures, toxicity from salts, etc ...), while this work shows that the rootstock more resistant to salinity (1103 P) has the highest F0 values. How can this be explained?

It could be under soil salinity F0 increased with 1103 Paulson rootstocks (resistance to soil senility) affected on proline and glycine content by which impact on the performance electron transport from Photosystem I to PII

  • Line 110 reports “version”, but it is correct to report “verasion”. Beware of similar errors in the rest of the text, the autocorrect makes these kinds of mistakes!

Corrected in the whole manuscript

  • What are the units of measurement in table 2?

mg 100 g-1 FW was added

  • The sentences from lines 116-126 appear somewhat confused and contradictory. Show clearer in the sentence At line 121 e 126 are reported the same data and among other things it is not clear what the data 1.33 vs 2.45 refer to 

  • The rulings between lines 127 and 133 are also very confusing and unclear.

127 was formatted

133 delated

  • In line 134 there is a reference to a Figure 3. Correctly Figure 2 should be indicated before Figure 3, however Figure 3 just does not exist in the text. How is it possible to comment on a figure that does not exist?

Sorry for that I main Table 3

  • Last consideration regarding the analysis of the pigments: it seems strange that these analyzes, always carried out on the leaves of the seventh node, show a progressive increase of chlorophyll during the season. Normally later on the seasion (at harvest time) leaf chlorophyll decrease.

Thank you for your question

Why in this experiment increase? Do the authors have explanations for this phenomenon?

Measurement of pigments increased in a conducted experiment may refer to the farm management programe applied for fertilization, especially foliar micronutriaties. However, the increases in pigment with rootstoces are different, which is why we applied chlorophyll fluresence parameters to clarifay the results. Although the increases in the chlorophyll fluresence parameters (Fm and F0) and pigments (cha and Car) remained two issues, Fv/Fm decreased as a result of rootstock effects we found a difference, and the increases in pigments for SO4 and own root rootstocks are insufficient to ensure a good yield under salinity conditions. Also, we observed that the levels of proline and glycine are different. In a comparison with 1103 Paulson rootstock, the results are different. Even we don't observe a completely salinity injured vines (In all leaves) 

This data is contradictory with the data on the SI (SS) index:

The leaf damage from salt toxicity increases with the season, but inexplicably the concentration of chlorophyll in the leaves also increases.

Of course, the experiment stopped after harvesting, but I need to know that all vines grown in sand need water (fertigation) every day, especially at a temperature of around 35 degrees under the sun. In unpublished experiment we found the foliare magnisum is active to treating the damage with sailinty damage up to normal leaves fall.

  • In conclusion, the comment on the data in chapter “3. Photosynthetic pigment (Chlorophyll and carotene content)”, must be substantially revised

Corrected in the whole manuscript

  • Line 146 reports “version”, but it is correct to report “verasion”. Beware of similar errors in the rest of the text, the autocorrect makes these kinds of mistakes!

Corrected in the whole manuscript

  • Line 161 reports “version”, but it is correct to report “verasion”. Beware of similar errors in the rest of the text, the autocorrect makes these kinds of mistakes!

Corrected in the whole manuscript

  • Line 166 reports “version”, but it is correct to report “verasion”. Beware of similar errors in the rest of the text, the autocorrect makes these kinds of mistakes!

Corrected in the whole manuscript

  • Re-tuning on M&M, in lines 409-410 it was not reported if the leaf area (cm2) is the average of the area of the single leaf or a shoot or of a vine. From the size it is assumed to be the area of ​​a single leaf. We formatted this part to be clear
  • If the surface is relative to a single leaf and if the 7th position leaf (node) has always been sampled, it is difficult to understand how this surface has increased during BDS as reported in lines 176-179. Can the authors explain this phenomenon?

Leaf samples were taken to estimate the leaf area as previously indicated on whole leaves, which are usually one month old.

  • Line 185: proline and glycine accumulation, are reported in Table 3 non in Table 4

Corrected

  • In lines 187-188 the sentence is appropriate for Discussion section not for Result ones

Transferred

  • In lines 201-202 it is reported: “Under this condition, Na+ increased while K+ and the k+/Na+ ratio decreased”. What conditions are referred to? The sentence is not clear.

I main the soil salinity changes to this meaning

  • Lines 202-204 report: “The '1103 Paulson' rootstock applied 202 resulted in a decreased Na+ content and increase in K+ concentration, and the K+/Na+ ratio 203 up to the harvest stage” . Why, the data are not as interesting: the self-grafted vines have high values ​​of Na and low of K in the leaves and above all very low values ​​of the K / Na ratio. Intermediate data between 1103 and Own root presented by SO4

I don’t know, but this is the result we got.

  • Line 207 refers to: “foliar fertilization at harvesting time”. Is the first time that foliar fertilization is mentioned, has it been done? When? Why is there no mention in M&M? Perhaps the term fertilization has been used to mean nutrition? However, the two terms are not synonymous.

It is mistakes in changes to vine rootstocks

  • Line 212 reports “version”, but it is correct to report “verasion”. Beware of similar errors in the rest of the text, the autocorrect makes these kinds of mistakes!
  • Corrected in the whole manuscript
  • Line 227 reports “version”, but it is correct to report “verasion”. Beware of similar errors in the rest of the text, the autocorrect makes these kinds of mistakes!
  • Corrected in the whole manuscript
  • Line 232 reports “version”, but it is correct to report “verasion”. Beware of similar errors in the rest of the text, the autocorrect makes these kinds of mistakes
  • Corrected in the whole manuscript
  • I propose that in table 7 the TA (lines 242-246) is expressed in g / L (per thousand) and not% as reported.

Total acidity was measured as a percentage

  • In line 257-258 was reported: All variables except IL percent (EL%?), MDA, and Na percent are negatively correlated with the SI- index. Considering that many of these correlations have very low 'r' and that vice versa the positive correlations are very high, I believe that the correct sentence should like: The variables EL%, MDA, and Na+ percent are positively correlated with the SI- index. While all the other variables are negatively correlated, even if many in an insignificant way, with SI.

Agreed with you

  • The sentences on lines 258-263 could be written more clearly. In table 4 the increase of EL% in own root vines is very interesting. Why is it not evidenced in lines 192-193?

Sorry for that, thank you...the description results was added.

  • The sentence on line 263: “Our observations were corroborated [21,22]” is appropriate for the Discussion chapter not for Results.

Transferred to discussion

Discussion

  • The sentence on lines 301-303 appears incorrect. Fv/Fm varies throughout the season, but its variation is independent of BDS. The decrease in Fv / Fm is due to normal leaf senescence, which runs parallel to BDS.

Formatted

  • As reported in the remarks on the Results chapter: Chla and Chlb's behavior during BDS appear to be strange. In all the rootstock combination, Chl increase during BDS. In all combinations of rootstock, Chla and Chlb increase during BDS. Normally they decrease with leaf senescence. Could the authors discuss these observed behaviors?

It was explained

  • The sentences in lines 308-318 seem to be unclear. What do these sentences mean?

Formatted

  • On lines 319-321 was reported: “ …and a reflection on the efficiency of the photosynthesis process that is not related to the amount of chlorophyll [37]”. But the 1013 rootstock increase both, in the same time, Fv/Fm and Chl.

Formatted

  • Also, the sentence on lines 325-327 seem to be unclear. “….the rootstocks used showed different soil salinity behaviors” (?) In is better: “In addition, the rootstocks used showed different response to the soil salinity content”

Formatted

  • On line 339 IL% means EL%?

Yes, changed

  • On line 339 what is meant by MG (never mentioned before)?

Changed,  it mean magnesium

  • On line 349 “root zoon” means “root zone”?

Yes correct

  • On line 351 what means “both potassium”, the sentence does not appear complete.

Delated word both

Final consideration

The work seems to have been carried out carefully and the results, even if limited to a few rootstocks, appear interesting and worthy of publication.

I propose a series of additions and corrections in the text.

It was a long and demanding job, but I hope it will be useful to the authors. Certainly, something will have escaped me. I am sure that another independent review will find other aspects that need to be corrected and improved.

My knowledge of English does not allow me to evaluate the quality of the text well, but I take the liberty of advising a careful revision by a native speaker who in any case knows well the topics covered.

Thank you very much, and I appreciate you effort

30 September 2021

Round 3

Reviewer 2 Report

I see that the authors have answered all my numerous questions and corrected the required points.  Therefore I give my approval to the publication in the final version.